# Predictors of alcohol use disorder risk in young adults: Direct and indirect psychological paths through binge drinking

Maxime Mauduy[1]*, Pierre Maurage[2], Nicolas Mauny[3], Anne-Lise Pitel[4], Hélène Beaunieux[5], Jessica Mange[5]

**1** Laboratoire de Psychologie Sociale (LPS, UR 4471), University of Paris Cité, Boulogne-Billancourt, France, **2** Louvain Experimental Psychopathology Research Group (LEP), Psychological Science Research Institute, UCLouvain, Louvain-la-Neuve, Belgium, **3** Laboratoire de Psychologie (UR 3188), University of Franche-Comté, Besançon, France, **4** Physiopathology and Imaging of Neurological Disorders, University of Caen Normandy, INSERM, Caen, France, **5** Laboratoire de Psychologie Caen Normandie (LPCN, UR 7452), University of Caen Normandy, Caen, France

* maxime.mauduy@u-paris.fr

## Abstract

Alcohol-use disorders (AUD) risk is highly prevalent in university students, and is associated with both intraindividual (e.g., metacognitions, personality traits) and interindividual (e.g., social motives, drinking identity, drinking norms) psychological factors. Binge drinking (BD) also constitutes a widespread drinking pattern in youth, distinct from AUD risk and mainly predicted by interindividual factors. As BD is itself a risk factor for AUD, we tested a dual psychological path model to AUD risk, combining a direct path (including intra/interindividual factors independent from BD) with an indirect path (where interindividual factors increase AUD risk through BD). We assessed a large range of psychological factors predicting BD and AUD risk in an online survey among 2026 university students (Mage = 20.46; SD = 2.82; 67.42% of women). We tested the direct and indirect (via BD) effects of these psychological factors on three subdimensions of AUDIT (alcohol intake, dependence symptoms, and alcohol-related problems) through a multivariate mediation model using percentile bootstrapped estimates. Support for the dual-path model to risk of AUD emerged from the results, comprising a direct path mainly influenced by intra-individual factors unrelated to BD (e.g., coping motives, depression symptoms, and uncontrollability beliefs), and an indirect BD-mediated path mainly influenced by inter-individual factors (e.g., social motives, enhancement motives, drinking norms) through BD. This new dual-path conceptualization combining direct/intra-individual and indirect/inter-individual risk factors identifies key psychological determinants of AUD risk in youth and offers new prevention avenues for AUD risk.

**Data availability statement:** Data that support these research results are accessible on OSF at (https://osf.io/hf4my/).

**Funding:** The work reported was supported by RIN Tremplin Grant 19E00906 of Normandie Région (France). This research was funded by IReSP and the Aviesan Alliance as part of the call for research projects to combat addiction to psychoactive substances Grant IRESP-19-ADDICTIONS-03. The funders had no role in study design, data collection and analysis, decision to publish, or preparation of the manuscript.

**Competing interests:** The authors have declared that no competing interests exist.

## Introduction

The prevalence of Alcohol Use Disorder (AUD) risk among university students is alarmingly high, with more than half of this population scoring above the threshold of seven/eight on the Alcohol Use Disorders Identification Test (AUDIT) [1–4]. Despite this, an integrated model that delineates the key intra-individual and inter-individual psychological variables contributing to AUD risk, particularly in the context of binge drinking (BD), remains absent. BD, defined as a drinking pattern characterized by alternating episodes of heavy alcohol consumption and periods of abstinence [5], is widespread among university students (e.g., 65.1% of students in a French sample presented occasional or frequent BD) and a known risk factor for the development of AUD [6–8]. Moreover, the psychological mechanisms underlying AUD risk in university students appear closely tied to those associated with BD, suggesting that the progression to AUD in this population often begins with BD. However, alcohol dependence symptoms and its associated problems, two major indictors of AUD, are distinct constructs from heavy consumption levels like BD [9–11]. This raises the possibility that other psychological mechanisms, beyond those associated with BD, may contribute to AUD risk in this population. Thus, this study seeks to refine our understanding of the pathways leading to AUD risk among university students by investigating the relationships between psychological factors, BD, and AUD risk. We hypothesize that certain psychological factors, depending on whether they are intra-individual or inter-individual, may exert direct effects on AUD risk, while others influence AUD risk indirectly via BD.

### Inter- and intra-individual psychological factors associated with AUD risk

A substantial body of research has identified psychological factors associated with AUD risk in university students, which can be broadly categorized as either inter-individual or intra-individual [12,13].

Three major inter-individual factors have consistently been linked to AUD risk in young populations. First, *drinking social identity* - the extent to which individuals identify themselves as part of the social category of "drinkers" [14,15] - positively predicts alcohol use and AUD risk among youth [15–18]. Second, based on Cooper's motivational model [19], two external drinking motives, namely *social* (i.e., positive external motives such as drinking to enhance social interactions) and *conformity* (i.e., negative external motives such as drinking to avoid social censure or rejection) *motives* have emerged as strong predictors of alcohol use [16,20] and AUD risk [19,21–23]. Notably, higher levels of social motives are positively associated with AUD risk, while conformity motives are negatively related to AUD risk. Third, *drinking social norms* - individuals' beliefs about the acceptability and prevalence of alcohol use among peers [24], influence alcohol use [25] and AUD risk [26]. Specifically, more positive perceptions of peer drinking are associated with alcohol consumption and increased AUD risk.

Three key intra-individual factors also contribute to AUD risk in young people. First, according to Cooper's motivational model, two internal drinking motives, namely *enhancement* (i.e., positive internal motive such as drinking to elevate positive mood)

and *coping* (i.e., negative internal motives such as drinking to regulate negative affect) *motives* are associated with greater alcohol use and AUD risk in young people [12,16,20,27]. Second, *alcohol-related metacognitions* - schematic information that individuals hold about the significance of their cognitive experience and ways to control it - are significant predictors of AUD risk [28,29]. These metacognitions include positive beliefs about the impact of alcohol use on *cognitive* (helping to control thoughts) and *emotional* (helping to improve mood) *self-regulation*, and negative metacognitions related to the perceived *uncontrollability* and the *negative cognitive functioning impact* of alcohol use. Third, *personality traits and psychopathological variables*, such as loneliness (i.e., a distressing feeling of isolation perception or social rejection [30,31]), anxiety [32], depression symptoms [33], and impulsivity (i.e., the tendency to act prematurely without considering consequences) have been consistently associated with alcohol use [34,35] and AUD risk [31,36,37]. More precisely, impulsivity, as conceptualized in the UPPS model [38], encompasses dimensions such as *negative* and *positive urgency* (i.e., the tendency to act rashly in intense negative or positive emotional contexts), *lack of perseverance* (i.e., the difficulty to maintain attention on a task), *lack of premeditation* (i.e., the difficulty to plan behaviors and consider their consequences), and *sensation seeking* (i.e., the propensity to look for new and exciting experiences), all of which have been linked to alcohol-related outcomes [39].

However, it is worth noting that these psychological determinants of AUD risk, and particularly drinking social identity and enhancement motives, are also the strongest predictors of BD [12,26], suggesting that BD and AUD may share common psychological predictors, that an *indirect psychological pathway* might lead to AUD through BD, and that preventing BD could reduce AUD risk among students.

## An indirect psychological pathway from BD to AUD

Social factors are widely recognized as primary initiators of alcohol use in young adults [40–43], a highly social behavior [44], characterized by the BD practice, a socially motivated drinking pattern [45]. Indeed, theoretical models, such as Koob and Volkow's three-stage conceptualization of addiction [46] and the socioecological framework of drinking contexts [47], emphasize that excessive alcohol consumption, and particularly BD, is a critical precursor to AUD development. Consistent with these models, research shows that BD is a significant risk factor for AUD [6,7,48].

BD, as a social drinking pattern, is primarily driven by psychological factors that can be classified into inter-individual (e.g., drinking social identity, drinking norms, social motives) and intra-individual (e.g., positive enhancement motives, sensation seeking) categories. These factors align with the positive affect regulation theory of AUD etiology, which conceptualizes AUD as a maladaptive strategy to enhance positive emotions and social experiences [40]. According to this theory, positive alcohol expectancies - beliefs about alcohol's ability to improve emotional states and facilitate social bonding - encourage individuals to consume alcohol and increase the likelihood of developing AUD. Supporting this perspective, studies have demonstrated that these BD-related psychological factors, especially interpersonal ones, are indirectly linked to AUD symptoms through higher levels of alcohol consumption [21,49,50].

Consequently, rather than exerting a direct influence on AUD risk, we hypothesize that these psychological factors primarily promote BD, which in turn increases AUD risk in young adults. In other words, psychological factors associated with the positive affect regulation conceptualization might lead young adults to engage in BD, which subsequently increases their AUD risk. However, existing studies often fail to account for this indirect pathway - and therefore report direct associations - due to two key methodological limitations.

## Methodological limitations affecting the understanding of AUD risk

First, research investigating psychological determinants of AUD typically examines these factors in isolation from BD, focusing on their direct effects on AUD risk without considering the mediating role of BD. This approach overlooks the possibility that BD may serve as a mediator between some psychological factors (e.g., drinking identity, social motives) and AUD risk [21,50], potentially explaining why many determinants appear common to both BD and AUD risk.

Second, most studies predicting AUD risk among university students use the overall AUDIT score as the outcome variable. According to the original authors, AUDIT measures three major dimensions associated with AUD risk: alcohol intake, alcohol dependence symptoms, and alcohol-related problems [51]. However, these three sub-dimensions (i) correspond to different aspects of alcohol consumption [9,10,51,52], (ii) have different predictors and causal factors [53], and (iii) are themselves different predictors of drinking behavior and its consequences [54]. Notably, the "alcohol intake" dimension is highly correlated with BD, as evidenced by the sensitivity and specificity of AUDIT-C in detecting BD [55,56]. In contrast, the other two dimensions - dependence symptoms and alcohol-related problems - capture distinct aspects of AUD risk from excessive consumption [9,10]. Consequently, studies relying on the overall AUDIT score may overestimate the role of BD-related psychological determinants in AUD risk and underestimate factors more closely tied to dependence symptoms and alcohol-related problems. Therefore, a growing body of evidence suggests that the AUDIT is not a unitary scale but should rather be distinguished according to its sub-dimensions to investigate the risk of AUD [9–11,57,58].

Thus, these two methodological issues may lead to (i) an overestimation of AUD predictors that are shared with BD but primarily act indirectly through BD, and (ii) an underestimation of predictors less correlated with BD but directly influencing AUD risk. Indeed, when focusing specifically on dependence symptoms and alcohol-related problems, some evidence suggests that specific psychological factors are associated with AUD risk among youth, highlighting a direct psychological pathway to AUD, independent from BD.

### A direct psychological pathway to AUD risk

One of the most prominent theories of AUD etiology suggests that AUD develops as a way to alleviate negative affect, such as reducing anxiety, stress, and depression [27,59–62]. Many young adults report drinking to cope with negative emotions [26,63,64]. Psychological factors associated with this negative affect regulation process - such as negative reinforcement motives (i.e., coping motives), negative alcohol-related metacognitions, and specific personality traits - may play a direct role in AUD risk, independent of BD, which would be less influenced by these factors. Empirical evidence supports this perspective. Studies focusing on the prediction of dependence symptoms and alcohol-related problems - rather than overall AUDIT scores - reveal notable differences between psychological predictors of BD and those of AUD risk.

Regarding drinking motives, while BD is primarily explained by social and enhancement motives, coping motives are more strongly associated with alcohol-related problems [27,65] and dependence symptoms [32,66]. Thus, coping motives appear to exert a direct influence on AUD risk, independent of BD practice.

Regarding alcohol-related metacognitions, empirical research highlights distinct roles for positive and negative metacognitions. Positive metacognitions, such as emotional self-regulation, are primarily associated with BD [28]. Conversely, negative metacognitions (e.g., beliefs about uncontrollability and negative consequences of alcohol use) are crucial for the persistence of alcohol consumption [67,68] and are significant predictors of alcohol dependence [69]. These findings suggest that negative metacognitions play a direct role in AUD risk among university students, independent of BD practice.

Regarding personality traits, the predictive role of impulsivity dimensions differs between alcohol use and alcohol-related problems or dependence symptoms. While the lack of perseverance is more strongly associated with alcohol use, negative urgency and the lack of premeditation are more predictive of AUD risk [39]. Additionally, loneliness is a characteristic specific to higher AUD risk compared to lower dependence risk [31], and depression symptoms have been more strongly associated with alcohol dependence symptoms than with general alcohol use [33]. These findings indicate that some impulsivity dimensions (e.g., negative urgency and lack of premeditation), as well as loneliness and depression, may directly contribute to AUD risk among university students, beyond their association with BD.

### The present research

The direct psychological determinants of AUD risk in university students, particularly concerning its dimensions of dependence symptoms and alcohol-related problems, may differ in nature from those of BD. We propose that the psychological

factors commonly associated with both AUD and BD risk in prior research may primarily reflect the mediating role of BD itself as a risk factor for AUD - a relationship often unaccounted for in existing statistical models. To address this gap, we hypothesize a dual-pathway model of AUD risk (see Fig 1), wherein:

(1) psychological factors less correlated with BD (e.g., coping motives, negative metacognitions, impulsivity dimensions, loneliness, and depression), mainly intra-individual in nature, exert a direct influence on AUD risk, independent of BD;

(2) psychological determinants of BD (e.g., drinking identity, social norms, social motives, enhancement motives, and sensation seeking), mainly inter-individual in nature, contribute indirectly to AUD risk, through the mediation of BD.

The first direct psychological pathway thus reflects the negative affect regulation conceptualization of AUD, while the second indirect psychological pathway primarily reflects the positive affect regulation conceptualization of AUD. Establishing this dual-pathway model could be critical for developing effective reduction strategies for AUD risk among university students by providing a nuanced understanding of the psychological mechanisms underlying AUD risk among youth. Accordingly, we tested this dual-pathway hypothesis in a large sample of university students.

## Method

For more details on the method, see the supporting information.

### Procedure and participants

We contacted 27,668 students from the university of Caen Normandy (France) through their institutional e-mail address to take part in an online survey (via the Limesurvey® application, from the beginning to the end of November 2019). The only inclusion criterion for taking part in the study was to be a Caen Normandie University student of legal age (18). We obtained 3,939 responses, this response rate (14.2%) being in line with previous studies in this population [70–73]. We then selected 2,026 eligible answers (i.e., students drinking alcohol regularly and having answered to the items related to alcohol consumption and psychological variables) which constituted the sample of the present study (see Fig 2 for the flow diagram and Table 1 for sample characteristics). Our sample is comparable in terms of socio-demographic data, alcohol consumption and psychopathological variables to those of previous studies exploring alcohol consumption among European university students [1,26,28,73–78]. This study was included in a larger research project exploring substance consumption among young adults (ADUC project: "Alcool et Drogues à l'Université de Caen").

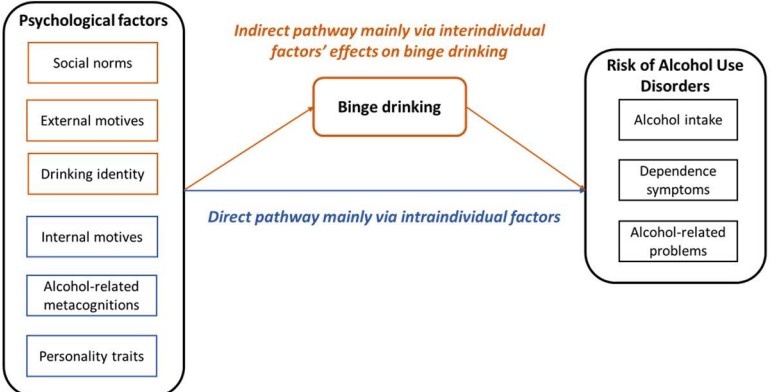

**Fig 1. The hypothesis of a psychological dual-path model of Risk of Alcohol Use Disorders in University students.**

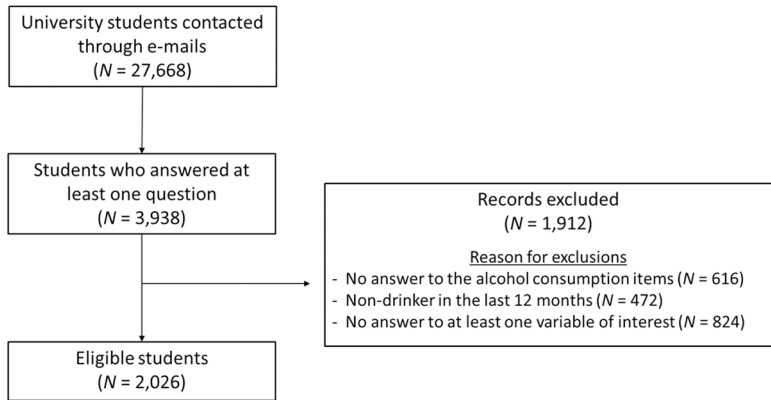

**Fig 2. Flow diagram.**

## Ethics

All participants gave their written consent before starting the survey. The study was notified to and authorized by the "Commission Nationale de l'Informatique et des Libertés" with the registration number u24- 20171109-01R1. This survey was conducted in full agreement with the Declaration of Helsinki (2008) and the ethical standards set by the Psychology Department, that follows the American Psychological Association Ethical Principles of Psychologists and the Code of Conduct (APA, 2019) for the ethical treatment of human participants.

## Measures

As our measures use four, five or seven points Likert-type scales, we have better assessed the reliability of the measures by using an ordinal coefficient alpha, rather than the classic Cronbach alpha [79].

**Sociodemographic variables.** We measured participants' gender and age.

**Alcohol consumption variables.** We computed a *BD score* using three questions (i.e., Q1: "number of average standard drinks (corresponding to 10 gr of ethanol in France) per hour", Q2: "number of times being drunk in the previous 12 months" and Q3: "percentage of times getting drunk when drinking" [12,80]. The computed BD score (i.e., [4 × Q1] + Q2 + [0.2 × Q3]) [81] allows us to consider both quantity and frequency of BD [45,81].

We measured risk of AUD from the Alcohol Use Disorders Identification Test (AUDIT[52]). The AUDIT is a 10-item measure designed to identify individuals at risk for AUD by assessing three subdimensions [52,66]: *alcohol intake* (items 1–3, frequency of drinking, number of drinks consumed on a typical day, frequency of heavy drinking), *alcohol dependence symptoms* (items 4–6, impaired control over drinking, increased salience of drinking, morning drinking), and *alcohol related-problems* (items 7–10, guilt after drinking, blackouts, alcohol-related injuries, others concerned about drinking). We tested the dimensionality of the AUDIT using confirmatory factor analyses. While the model comprising a single dimension (overall AUDIT score) showed a poor fit to the data ($\chi^2_{(35)}$ = 1128, $p < .001$, CFI = .852, TLI = .810, RMSEA = .105, 90% CI [.10,.11], SRMR = .049), the model distinguishing the three AUDIT sub-dimensions showed an excellent fit to the data ($\chi^2_{(32)}$ = 383, $p < .001$, CFI = .953, TLI = .933, RMSEA = .062, 90% CI [.057,.068], SRMR = .035). This reinforces the idea of avoiding the use of an overall AUDIT score to rather measure each of the three AUDIT subdimensions separately to investigate AUD risk. Thus, we computed a score for alcohol intake ($\alpha = 0.73$), a score for alcohol dependence symptoms ($\alpha = 0.81$), and a score for alcohol-related problems ($\alpha = 0.76$).

**Inter-individual psychological predictors.** *Drinking identity,* that is the extent to which alcohol use is important to define the participant's identity, was assessed with a 2-item Likert-type scale from 1 = *do not agree* to 4 = *agree very much*; $\alpha = 0.83$ [12,82].

**Table 1. Sample characteristics.**

| | | M (SD) | Range |
|---|---|---|---|
| **Socio-demographics** | | | |
| | Females, N (%) | 1366 (67.42%) | |
| | Males, N (%) | 660 (32.58%) | |
| | Age, years | 20.46 (2.82) | 18 - 35 |
| **Alcohol consumption variables** | | | |
| | Binge drinking score | 20.46 (21.71) | 1.33 - 246.60 |
| | AUDIT total score | 7.08 (5.49) | 1 - 32 |
| | Alcohol intake | 4.20 (2.29) | 1 - 12 |
| | Alcohol dependence symptoms | 0.86 (1.50) | 0 - 12 |
| | Alcohol related-problems | 2.02 (2.73) | 0 - 16 |
| **Inter-individual psychological variables** | | | |
| | Drinking identity | 1.50 (0.98) | 1 - 7 |
| | Drinking social norms | 2.11 (1.31) | 1 - 7 |
| | DMQ-R - Social motives | 8.58 (3.41) | 1 - 15 |
| | DMQ-R - Conformity motives | 4.60 (2.37) | 3 - 15 |
| **Intra-individual psychological variables** | | | |
| | DMQ-R - Enhancement motives | 8.32 (3.35) | 3 - 15 |
| | DMQ-R - Coping motives | 5.64 (2.97) | 1 - 15 |
| | PAMS - Emotional self-regulation | 21.0 (5.89) | 7 - 32 |
| | PAMS - Cognitive self-regulation | 5.36 (1.72) | 3 - 16 |
| | NAMS - Uncontrollability | 3.29 (0.85) | 1 - 10 |
| | NAMS - Cognitive harm | 6.21 (2.58) | 1 - 12 |
| | STAI-T - Anxiety | 48.01 (11.95) | 2 - 80 |
| | BDI - Depression symptoms | 6.92 (6.21) | 0 - 38 |
| | ESUL - Loneliness | 35.69 (11.28) | 10 - 77 |
| | UPPS - Negative urgency | 9.04 (2.92) | 4 - 16 |
| | UPPS - Positive urgency | 10.62 (2.64) | 4 - 16 |
| | UPPS - Lack of premeditation | 7.42 (2.31) | 2 - 16 |
| | UPPS - Lack of perseverance | 7.50 (2.54) | 1 - 16 |
| | UPPS - Sensation seeking | 10.01 (2.94) | 3 - 16 |

Note. N = 2,026. Except for sex, data show means (standard deviations); ESUL: Echelle de Solitude de l'Université de Laval; STAI-T: State-Trait Anxiety Inventory; UPPS: Impulsive Behavior Scale; BDI: Beck Depression Inventory; AUDIT: Alcohol Use Disorders Identification Test; PAMS & NAMS: Positive and Negative Alcohol Metacognitions Scales; DMQ-R: Drinking Motives Questionnaire Revised.

*Drinking social norms*, namely how much most of the participants' significant relatives approve and/or adopt alcohol consumption to "get drunk", were assessed with a 2-item Likert-type scale from 1 = *do not agree* to 7 = *agree very much* and one item from 1 = *no person* to 6 = *5 persons* (α = 0.89) [12].

*Drinking motives* are the individuals' reasons for engaging in alcohol use and were assessed with the Drinking Motives Questionnaire (DMQ-R) [83]. It is a 12-item scale rated on a 5-point scale ranging from 1 = *never* to 5 = *always*, including, as interindividual motives, social (α = 0.90) and conformity (α = 0.90) subscales, and as intraindividual motives, coping (α = 0.92), enhancement (α = 0.85) subscales.

**Intra-individual psychological predictors.** *Alcohol-related metacognitions* were assessed using the French version of the Positive Alcohol Metacognitions Scale (PAMS) and the Negative Alcohol Metacognitions Scale (NAMS; Likert-type scales from 1 = *do not agree* to 4 = *agree very much*) [28]. We measured metacognitions about emotional (i.e., drinking

 

helps to improve mood; 8-item, α = 0.92) and cognitive (i.e., drinking helps to control thoughts; 4-item, α = 0.80) self-regulation and uncontrollability (i.e., perceived loss of control due to alcohol use; 3-item, α = 0.76) and cognitive harm (i.e., perceived negative cognitive functioning impact of alcohol use; 3-item, Cronbach α = 0.81).

*Impulsivity* was measured using the French short version of the UPPS-P Impulsive Behavior Scale [84]. It is a 20-item Likert-type scale ranging from 1 = *do not agree* to 4 = *agree very much*. We measured five facets of impulsivity, namely positive (α = 0.81) and negative (α = 0.85) urgency (i.e., the tendency to act rashly to regulate negative and positive emotions), lack of premeditation (i.e., the tendency to act without thinking; α = 0.86), lack of perseverance (i.e., the tendency to not finish tasks; α = 0.92), and sensation seeking (i.e., the tendency to seek out new or thrilling experience; α = 0.86).

*Anxiety* was measured with the French version of the State-Trait Anxiety Inventory (STAI) [85]. It is a 20-item scale ranging from (1) *no* to (4) *yes* (α = .94).

*Depression symptoms* were assessed with the short French version of the Beck Depression Inventory (BDI) [86]. It is a 13-item scale where participants answer (1) *yes* or (0) *no*.

*Loneliness* was measured with the ESUL (i.e., "Echelle de Solitude de l'Université de Laval"), a Canadian-French speaking adaptation of the UCLA Loneliness Scale [87]. It is a 20-item scale ranging ranging from 1 = *never* to 4 = *often* (α = 0.95).

## Statistical analysis

We conducted a multivariate mediation model with psychological factors as predictors, BD behavior as the mediator, and the three subdimensions of AUDIT as outcomes.

To account for gender effects, we included gender as a covariate in the model tested (coded as Female: -0.5; Male: 0.5). Gender was significantly associated with BD behavior ($\beta$ = .10, 95% CI [.07,.14]), drinking identity ($\beta$ = .18, 95% CI [.14,.23]), social motives ($\beta$ = .12, 95% CI [.08,.17]), enhancement motives ($\beta$ = .07, 95% CI [.03,.11]), conformity motives ($\beta$ = -.05, 95% CI [-.09, -.007]), drinking norms ($\beta$ = .11, 95% CI [.06,.15]), negative urgency ($\beta$ = -.15, 95% CI [-.19, -.11]), positive urgency ($\beta$ = -.09, 95% CI [-.13, -.04]), lack of perseverance ($\beta$ = .07, 95% CI [.03,.11]), sensation seeking ($\beta$ = .15, 95% CI [.11,.19]), anxiety ($\beta$ = -.26, 95% CI [-.30, -.22]), depression symptoms ($\beta$ = -.15, 95% CI [-.19, -.11]), emotional self-regulation ($\beta$ = -.06, 95% CI [.01,.10]), cognitive self-regulation ($\beta$ = .06, 95% CI [.01,.10]), and cognitive harm ($\beta$ = -.04, 95% CI [-.09, -.001]).

Given the non-normality observed in several variables - common in drinking behavior data due to positive skewness and/or kurtosis [88] - we applied a $\log_{10}$ transformation to the following variables: cognitive self-regulation, uncontrollability beliefs, drinking identity, conformity motives, binge drinking, and alcohol dependence symptoms. To further address potential violations of multivariate normality, we utilized maximum likelihood estimation combined with non-parametric percentile bootstrapping ($N$ = 10,000 resamples) to estimate path coefficients and 95% Confidence Intervals (CIs). This bootstrapping approach effectively mitigates issues related to non-normal data distributions [89,90]. As the bootstrapping approach does not allow us to deal with missing data, our statistical model carried out on our sample of 2,026 students did not include any missing data.

The model tested both direct effects of the psychological predictors on the AUDIT subdimensions (alcohol intake, dependence symptoms, and alcohol-related problems) and their indirect effects mediated by BD. To determine the presence of mediation effects, we adhered to the criteria outlined by Yzerbyt et al. [91], which require that: the effect of the predictor on the mediator (*a*), the effect of the mediator on the outcome (*b*), and the indirect effect (*a*b*) must all be significant. All statistical analyses were performed using R software (version 4.05), specifically employing the *mvn* [92] and *lavaan* [93] packages.

## Results

Fig 3 illustrates the main results of the analyses.

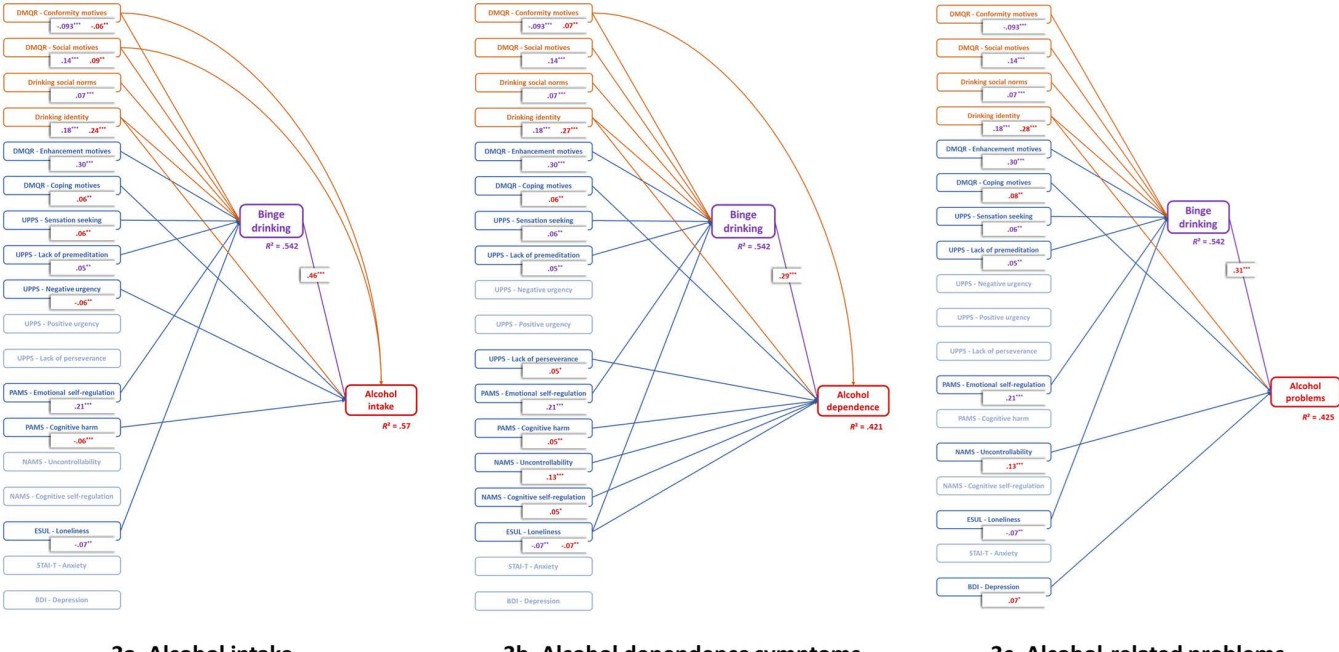

**3a. Alcohol intake**  **3b. Alcohol dependence symptoms**  **3c. Alcohol-related problems**

*Note.* Inter-individual factors are in orange. Intra-individual factors are in blue. Transparent factors are those that show no significant association with binge drinking or alcohol-related outcomes. Numbers below factors are standardized coefficients. Numbers in purple are for the link between factors and binge drinking. Numbers in red are for the links between factors and outcomes (alcohol intake, alcohol dependence, and alcohol-related problems). Statistically significant at ***$p < .001$; **$p < .01$; *$p < .05$.

**Fig 3. Results of the psychological dual-path model of Risk of Alcohol Use Disorders in University students according to the three AUDIT dimensions.**

## Direct predictors of AUDIT subdimensions

We identified a direct pathway to AUD risk, predominantly influenced by intraindividual predictors (see Table 2). As expected, beyond the direct effect of BD, alcohol dependence symptoms and alcohol-related problems are more predicted by intraindividual factors - such as coping motives, metacognitions related to cognitive self-regulation, uncontrollability beliefs and cognitive harm, lack of perseverance, loneliness, and depression symptoms - than by interindividual factors, which include drinking identity and conformity motives. In contrast, alcohol intake was associated with both interindividual factors (i.e., drinking identity, conformity motives, and social motives) and intraindividual factors (i.e., coping motives, cognitive harm, and negative urgency). Additionally, other factors, such as enhancement motives and drinking norms, did not show a direct association with any of the AUDIT subdimensions.

## Indirect predictors of AUDIT subdimensions through BD

Results indicated that BD is mostly predicted by inter-individual factors including drinking identity, social motives, enhancement motives, conformity motives, and drinking social norms, but also by some intraindividual factors, such as emotional self-regulation, lack of premeditation, sensation seeking, and loneliness (see Table 2). Furthermore, nearly all of these direct predictors of BD demonstrated significant indirect effects on the three AUDIT subdimensions through BD behavior, with the exception of loneliness and lack of premeditation, which did not significantly influence alcohol dependence symptoms (see Table 3).

 

**Table 2. Bootstrap analyses of direct effects for factors predicting the three subdimensions of AUDIT and binge drinking.**

| Outcomes | Alcohol intake ($R^2 = .570$) | | Alcohol dependence symptoms ($R^2 = .421$) | | Alcohol-related problems ($R^2 = .425$) | | Binge drinking ($R^2 = .542$) | |
|---|---|---|---|---|---|---|---|---|
| **Psychological variables** | β | 95% CI | β | 95% CI | β | 95% CI | β | 95% CI |
| Binge drinking | **.46***** | **[.42,.50]** | **.29***** | **[.24,.33]** | **.31***** | **[.26,.36]** | / | / |
| *Inter-individual variables* | | | | | | | | |
| Drinking identity | **.24***** | **[.20,.28]** | **.27***** | **[.21,.32]** | **.28***** | **[.23,.34]** | **.182***** | **[.15,.22]** |
| Drinking social norms | -.02 | [-.06,.01] | .01 | [-.04,.05] | -.01 | [-.05,.04] | **.073***** | **[.04,.11]** |
| DMQ-R - Social motives | **.09**** | **[.04,.15]** | .03 | [-.04,.09] | .05 | [-.02,.11] | **.139***** | **[.09,.19]** |
| DMQ-R - Conformity motives | **-.06**** | **[-.10, -.02]** | **.07**** | **[.03,.11]** | .02 | [-.03,.06] | **-.093***** | **[-.13, -.06]** |
| *Intra-individual variables* | | | | | | | | |
| DMQ-R - Coping motives | **.06**** | **[.02,.10]** | **.07*** | **[.02,.12]** | **.08**** | **[.03,.14]** | .031 | [-.01,.07] |
| DMQ-R - Enhancement motives | .05 | [-.00,.11] | .02 | [-.04.08] | -.03 | [-.09,.04] | **.295***** | **[.24.35]** |
| PAMS – Emotional regulation | .00 | [-.04,.04] | -.03 | [-.02,.08] | -.04ᵗ | [-.09,.00] | **.206***** | **[.16,.25]** |
| PAMS – Cognitive regulation | .04 | [-.00,.07] | **.05*** | **[.01,.09]** | .05ᵗ | [-.01,.10] | -.031 | [-.07,.01] |
| NAMS – Uncontrollability | -.00 | [-.04,.03] | **.13***** | **[.08,.17]** | **.13***** | **[.08,.18]** | .032 | [.00,.06] |
| NAMS – Cognitive harm | **-.06***** | **[-.09, -.03]** | **.05**** | **[.01,.10]** | .00 | [-.03,.04] | .004 | [-.03,.03] |
| UPPS - Negative urgency | **-.06**** | **[-.09, -.02]** | .00 | [-.08,.04] | -.04ᵗ | [-.08,.01] | -.017 | [-.05,.02] |
| UPPS - Positive urgency | -.02 | [-.05,.02] | .03 | [-.02,.08] | .04ᵗ | [-.00,.09] | -.019 | [-.06,.02] |
| UPPS – Lack of premeditation | .03 | [-.00,.07] | .03 | [-.01,.08] | .05ᵗ | [-.00,.09] | **.045*** | **[.01,.08]** |
| UPPS - Lack of perseverance | -.00 | [-.04,.03] | **.05*** | **[.01,.09]** | .02 | [-.02,.06] | .023 | [-.01,.06] |
| UPPS – Sensation seeking | .03 | [-.00,.06] | .00 | [-.03,.04] | .03 | [-.01,.06] | **.057**** | **[.02,.09]** |
| Anxiety | -.015 | [-.07,.04] | -.02 | [-.08,.03] | -.01 | [-.07,.04] | -.018 | [-.07,.03] |
| Loneliness | -.04 | [-.08,.00] | **-.07**** | **[-.12, -.03]** | -.04ᵗ | [-.08,.00] | **-.069**** | **[-.11, -.03]** |
| Depression symptoms | -.03 | [-.08,.02] | .04 | [-.02,.10] | **.07*** | **[.01,.13]** | .041 | [-.01,.09] |

*Note. N = 2,026. Confidence intervals were derived using the percentile bootstrap method with 10,000 resamples. Statistically significant at ***$p < .001$; **$p < .01$; *$p < .05$. Gender variable was included as a covariate in the model.*

## Discussion

We aimed to deepen the understanding of the psychological determinants of AUD risk in university students by addressing two key gaps in previous research. First, we explored the possibility of a dual psychological pathway to AUD risk: a direct path, primarily driven by intraindividual factors, and an indirect path, mediated by BD - a known risk factor for AUD - predominantly involving interindividual factors. Second, we distinguished between the three subdimensions of the AUDIT - alcohol intake, alcohol dependence symptoms, and alcohol-related problems - since their direct psychological determinants may differ. Our study, therefore, investigated this dual psychological pathway to AUD risk in a sample of university students.

### Evidence of a dual pathway to AUD risk in university students

Our results first identified a direct pathway to AUD risk, independent of BD. Overall, AUD risk is predominantly predicted by intraindividual factors such as alcohol-related metacognitions and impulsivity, rather than by interindividual factors like drinking social norms. This direct intraindividual pathway aligns with existing theoretical frameworks and empirical findings, which emphasize the significant role of intraindividual and personality factors in AUD, particularly in relation to dependence symptoms and alcohol-related problems [32,39,66,68,94]. Moreover, we demonstrated that this direct pathway to AUD risk is distinct from the one leading to BD, which is primarily influenced by interindividual factors (e.g., drinking

**Table 3. Bootstrap analyses of indirect effects for psychological variables predicting the three subdimensions of AUDIT via binge drinking behavior.**

| Psychological predictor | Mediator | Outcome | b | 95% CI for unstandardized indirect effect | β | 95% CI for standardized indirect effect |
|---|---|---|---|---|---|---|
| Drinking identity | BD | Alcohol intake | .192 | [.151,.235] * | .083 | [.065,.101] |
| Social motives | BD | Alcohol intake | .043 | [.026,.060] * | .064 | [.039,.088] |
| Enhancement motives | BD | Alcohol intake | .093 | [.075,.111] * | .135 | [.109,.160] |
| Conformity motives | BD | Alcohol intake | -.098 | [-.137, -.060] * | -.043 | [-.059, -.026] |
| Drinking social norms | BD | Alcohol intake | .096 | [.049,.144] * | .034 | [.017,.050] |
| Lack of premeditation | BD | Alcohol intake | .021 | [.005,.037] * | .021 | [.005,.037] |
| Sensation seeking | BD | Alcohol intake | .020 | [.008,.033] * | .026 | [.010,.042] |
| Loneliness | BD | Alcohol intake | -.006 | [-.010, -.003] * | -.032 | [-.050, -.013] |
| Emotional self-regulation | BD | Alcohol intake | .217 | [.162,.272] * | .094 | [.070,.118] |
| | | | | | | |
| Drinking identity | BD | Alcohol dependence | .013 | [.010,.017] * | .052 | [.039,.064] |
| Social motives | BD | Alcohol dependence | .003 | [.002,.004] * | .040 | [.023,.056] |
| Enhancement motives | BD | Alcohol dependence | .006 | [.005,.008] * | .084 | [.063,.104] |
| Conformity motives | BD | Alcohol dependence | -.007 | [-.010, -.004] * | -.027 | [-.037, -.016] |
| Drinking social norms | BD | Alcohol dependence | .007 | [.003,.010] * | .021 | [.010,.032] |
| Lack of premeditation | BD | Alcohol dependence | .001 | [.000,.003] | .013 | [.003,.023] |
| Sensation seeking | BD | Alcohol dependence | .001 | [.001,.002] * | .016 | [.006,.027] |
| Loneliness | BD | Alcohol dependence | .000 | [-.001,.000] | -.020 | [-.031, -.008] |
| Emotional self-regulation | BD | Alcohol dependence | 0.015 | [.011,.019] * | .059 | [.042,.075] |
| | | | | | | |
| Drinking identity | BD | Alcohol problems | .153 | [.118,.192] * | .056 | [.043,.069] |
| Social motives | BD | Alcohol problems | .034 | [.021,.048] * | .043 | [.026,.060] |
| Enhancement motives | BD | Alcohol problems | .074 | [.057,.093] * | .091 | [.069,.113] |
| Conformity motives | BD | Alcohol problems | -.079 | [-.110, -.049] * | -.029 | [-.040, -.018] |
| Drinking social norms | BD | Alcohol problems | .077 | [.039,.119] * | .023 | [.011,.034] |
| Lack of premeditation | BD | Alcohol problems | .017 | [.004,.030] * | .014 | [.003,.025] |
| Sensation seeking | BD | Alcohol problems | .016 | [.006,.027] * | .018 | [.007,.029] |
| Loneliness | BD | Alcohol problems | -.005 | [-.008, -.002] * | -.021 | [-.034, -.009] |
| Emotional self-regulation | BD | Alcohol problems | .174 | [.126,.225] * | .064 | [.046,.082] |

*Note. N = 2,026. Confidence intervals (CI) were derived using the percentile bootstrap method with 10,000 resamples.*

*Indirect effect is statistically significant at *p* < .05 and 95% CI unstandardized does not include zero. Gender variable was included as a covariate in the model.

identity, social motives, drinking norms). This distinction supports previous research on BD determinants [12], reinforcing the idea that BD is fundamentally a positive reinforcement social drinking pattern [12,80], while AUD risk is more closely related to negative regulatory strategies [19,63,95–97], often involving negative reinforcement motives (e.g., coping motives, cognitive self-regulation) associated with negative thoughts (e.g., uncontrollability and cognitive harm beliefs) used to manage psychological distress (e.g., depression, loneliness).

Second, our findings reveal that traditional interindividual factors [12,66,98], such as social motives, conformity motives, and drinking social norms, are largely indirect determinants of AUD risk in university students. Their influence on AUD risk, particularly regarding alcohol dependence symptoms and alcohol-related problems, is mediated by BD. While these findings are consistent with previous research identifying BD as a risk factor for AUD [6,7], they underscore the existence of an indirect psychological pathway to AUD risk, mediated by BD. This research results do not challenge the notion that social factors, particularly social norms, play a significant role in AUD and alcohol-related problems. Extensive evidence demonstrates that social norms influence the negative consequences of alcohol consumption [99,100] and that prevention strategies targeting these norms can effectively reduce such consequences [101]. However, our results rather suggest that the influence of social norms may be more directly related to excessive alcohol consumption behaviors (i.e., BD) than to dependence symptoms or the associated negative outcomes, consistently with some previous studies [47,102]. Social factors may serve as a gateway to alcohol consumption among young people, encouraging heavy social drinking behaviors such as BD, which in turn lead to negative consequences. The risk of AUD among young people appears to arise indirectly, as their normative social environment promotes BD, thereby contributing to the emergence of dependence symptoms and alcohol-related problems along this indirect psychological pathway. Thus, this indirect pathway clarifies the role of interindividual factors and BD as predictors of AUD in university students.

Our results thus support the dual-pathway model to AUD risk in university students, with an indirect pathway via BD, primarily consisting of interindividual factors and accounting for 37.12% of alcohol intake, 20% of dependence symptoms, and 22.61% of alcohol-related problems, and a direct pathway, primarily consisting of intraindividual factors. This is in line with a recent conceptualization of AUD risk in young people that also considers two pathways to AUD [103]. The first pathway to AUD in young people would involve the social practice of alcohol consumption, predicted by psychosocial factors linked to the idea of alcohol consumption for positive reinforcement purposes. A second pathway to AUD in young people would involve solitary drinking, predicted by intra-individual psychological factors associated with the idea of drinking for negative reinforcement.

Therefore, future studies should further explore our dual pathway, particularly to identity the drinking patterns of youth who follow the direct pathway to AUD risk, and particularly the influence of solitary drinking. Additionally, it is important to determine whether these two pathways are mutually exclusive or whether they can coexist within the same individual. While our variable-centered statistical approach demonstrates the presence of both pathways as distinct and independent - each operating beyond the influence of the other - it also suggests that they may coexist. Supporting this notion, recent research suggests that these two pathways might operate simultaneously within individual. Youth solitary drinkers, who could instead take the direct path to AUD, still spend the majority of their time drinking in social settings [104,105]. Moreover, Lannoy et al. [20] identified a psychological profile of binge drinkers characterized by both positive reinforcement motives (e.g., social and enhancement) and negative ones (e.g., coping motives), while Lannoy et al. [71] identified a profile of alcohol users characterized by both positive and negative reinforcement motives (e.g., social and coping motives) and impulsivity traits (e.g., lack of premeditation and perseverance). These profiles were found to be at the highest risk of AUD compared to profiles characterized only by positive reinforcement motives (e.g., enhancement and social). Thus, both pathways might coexist in some students, amplifying their AUD risk, while others may only present the indirect pathway via BD. The existing literature has yet to identify a group of non-binge-drinking students characterized predominantly by intraindividual factors such as coping motives or personality traits. Thus, future research should better delineate this direct pathway to AUD risk in university students, including its relationship to consumption patterns that do not involve BD. Additionally, the hypothesis that the direct pathway may develop after the indirect one in some students warrants exploration through longitudinal studies, as it aligns with recent findings showing that drinking motives change over time and interact: social motives may increase coping motives, while coping motives may reduce enhancement motives [106]. Finally, although our research aimed to explore the major pathways of psychological factors that predict AUD risk, intra-individual and inter-individual factors may interact together to predict BD or AUD risk notably because each of these two alcohol outcomes is always influenced by a specific combination of inter-individual and intra-individual factors. Therefore, clarifying

the interactions between these two types of factors within each pathway would be an interesting research perspective to pursue by using cluster or latent profile analyses.

### Drinker identity as a key psychological factor in both pathways to AUD risk

Moreover, our results also underscore the significance of one key psychological factor: drinking identity. This factor uniquely predicts AUD both directly and indirectly through BD, consistent with previous studies on BD [12] and AUD [26]. This highlight the importance of drinking identity in preventive measures and encourage further investigation into its role in AUD. Drinking identity comprises two interconnected yet distinct aspects: social identity (i.e., viewing oneself as a member of a social category) [12] and personal identity (i.e., considering a behavior as a core part of the self) [97]. It is plausible that its dual role in predicting AUD risk directly and influencing it indirectly through BD reflects these two aspects: personal drinking identity for the direct pathway and social drinking identity for the indirect pathway to AUD risk.

### Practical implications of the dual pathway model of AUD risk

Since BD is a highly prevalent drinking pattern among students and a risk factor for developing AUD, practitioners might focus on preventing this drinking behavior to mitigate the risk of AUD. To achieve this, prevention interventions should target the key psychological determinants of BD, and centrally social factors. Interventions addressing social norms through techniques like normative feedback [107] or induced hypocrisy [108], as well as interventions targeting drinker social identity using the multi-categorization technique [109], could prove effective in reducing this excessive alcohol consumption pattern and reduce the transition towards AUD.

However, the findings of this research, particularly the identification of the dual pathway, suggest that BD is not the sole route to AUD risk among young people. Other psychological factors, primarily intra-individual in nature and only weakly linked (or unrelated) to BD, also contribute to AUD risk, and more particularly the symptoms of dependence and the negative consequences associated with alcohol consumption. Therefore, it is crucial not to overlook, in the context of alcohol consumption among students, prevention interventions targeting coping drinking motivations, alcohol-related negative metacognitive beliefs, impulsivity and depressive symptoms. For instance, interventions focusing on coping motives with a framing technique associated with short-term perspective [110,111], metacognitive beliefs via persuasive communication [112], impulsivity through cognitive remediation techniques [113], and depressive symptoms via mindfulness-based interventions [114] could be effective in mitigating this direct psychological pathway toward AUD in students.

Thus, the findings of this research suggest that, rather than prioritizing interventions targeting either the indirect pathway to AUD or its direct pathway, designing a comprehensive intervention program that combines both sets of interventions would likely be the most effective approach to counter AUD risk among students.

### Limitations and future directions

First, although previous longitudinal studies [66,115] support the direction of relationships posited in the dual-pathway model to AUD risk tested here, our study utilized a cross-sectional design. Second, the self-reported nature of the survey may have introduced recall and social desirability biases, though the anonymous nature of online surveys likely mitigated this bias to some extent. Third, we tested our dual-pathway model by considering all psychological factors at the same level of analysis. Future research should aim to refine this model by considering the interplay and mutual influences among psychological factors [27,98,115]. This single-level analysis may explain why some psychological factors (e.g., anxiety) known to influence alcohol use and AUD were not found to be significant in our study. Finally, while our results are consistent with previous studies [12,66] and while the model tested appears generalizable to other populations and contexts, beyond university students (since the variables tested are general determinants of alcohol consumption and AUD risk beyond specific contexts), their generalizability should be confirmed in future studies on more various samples.

## Conclusion

We developed and tested a dual psychological pathway to AUD risk. On the one hand, we identified a direct pathway, primarily involving intraindividual factors (e.g., coping motives and personality traits), which directly influence alcohol dependence symptoms and alcohol-related problems, distinct from those associated with BD. On the other hand, we demonstrated an indirect pathway, predominantly based on interindividual factors (e.g., drinking norms, social motives), which predicts AUD risk indirectly through BD. This study enhances the understanding of AUD risk predictors in university students by clarifying the respective roles of intraindividual and interindividual factors by taking into account the role of BD. Highlighting this dual psychological pathway holds significant implications for prevention efforts. Indeed, it emphasizes that attention should not be solely directed towards addressing the key psychological factors of BD alone, as AUD risk among students does not solely stem from BD practices and may involve psychological factors distinct from those associated with BD.

## Supporting information

**S1 File. Details of study measures.**
(DOCX)

## Author contributions

**Conceptualization:** Maxime Mauduy, Hélène Beaunieux, Jessica Mange.

**Data curation:** Maxime Mauduy, Nicolas Mauny.

**Formal analysis:** Maxime Mauduy.

**Funding acquisition:** Hélène Beaunieux.

**Investigation:** Maxime Mauduy, Nicolas Mauny.

**Methodology:** Maxime Mauduy, Pierre Maurage, Jessica Mange.

**Project administration:** Hélène Beaunieux, Jessica Mange.

**Resources:** Maxime Mauduy, Pierre Maurage, Anne-Lise Pitel, Jessica Mange.

**Software:** Maxime Mauduy.

**Supervision:** Pierre Maurage, Jessica Mange.

**Validation:** Maxime Mauduy.

**Visualization:** Maxime Mauduy.

**Writing – original draft:** Maxime Mauduy.

**Writing – review & editing:** Maxime Mauduy, Pierre Maurage, Nicolas Mauny, Anne-Lise Pitel, Hélène Beaunieux, Jessica Mange.

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
