## [Decision Letter · Decision Letter 0]

12 Nov 2024

PONE-D-24-39096Predictors of Alcohol Use Disorder Risk in Young Adults: Direct and Indirect Psychological Paths through Binge DrinkingPLOS ONE

Dear Dr. Mauduy,

Thank you for submitting your manuscript to PLOS ONE. After careful consideration, we feel that it has merit but does not fully meet PLOS ONE’s publication criteria as it currently stands. Therefore, we invite you to submit a revised version of the manuscript that addresses the points raised during the review process.

We look forward to receiving your revised manuscript.

Kind regards,

Sujiv Akkilagunta, M.D. Community Medicine

Academic Editor

PLOS ONE

“The work reported was supported by RIN Tremplin Grant 19E00906 of Normandie Région (France). This research was funded by IReSP and the Aviesan Alliance as part of the call for research projects to combat addiction to psychoactive substances Grant IRESP-19-ADDICTIONS-03.”

3. Please note that your Data Availability Statement is currently missing the DOI/accession number of each dataset OR a direct link to access each database. If your manuscript is accepted for publication, you will be asked to provide these details on a very short timeline. We therefore suggest that you provide this information now, though we will not hold up the peer review process if you are unable.

Additional Editor Comments:

The study is well conceived and conducted.

The non-response rate is on the higher side. Is there any effort made to assess if there is a systemic difference between responders and non-responders?

If there is a diagrammatic representation of the final pathway model, it will add to comprehensibility of the study.

The comments by the reviewers may be addressed.

Reviewers' comments:

Reviewer's Responses to Questions

**Comments to the Author**

1. Is the manuscript technically sound, and do the data support the conclusions?

Reviewer #1: Partly

Reviewer #2: Yes

2. Has the statistical analysis been performed appropriately and rigorously? 

Reviewer #1: Yes

Reviewer #2: I Don't Know

3. Have the authors made all data underlying the findings in their manuscript fully available?

Reviewer #1: Yes

Reviewer #2: Yes

4. Is the manuscript presented in an intelligible fashion and written in standard English?

Reviewer #1: Yes

Reviewer #2: Yes

5. Review Comments to the Author

Reviewer #1: Comment 1: Indicate the number of participants with missing data for each variable of interest

Comment 2: Although the introduction lists psychological determinants like drinking motives and personality traits, there is minimal operational definition of these constructs.

For instance, "impulsivity" or "drinking identity" could be elaborated upon to provide readers a better grasp of their theoretical and practical implications within this model.

Comment 3: The choice to focus on three AUDIT subdimensions—alcohol intake, dependence symptoms, and alcohol-related problems—is central to the study, yet the rationale for this distinction could be clearer. Further expounding on how these subdimensions uniquely relate to AUD risk and why it is important to differentiate them would strengthen the study's conceptual framework.

Comment 4: Although the dual-pathway model proposed is intriguing, a more detailed outline of the hypothesized mechanisms for each pathway (direct vs. indirect) would clarify the expected relationships. For instance, it is unclear if and how specific intra-individual factors might interact with inter-individual ones or if the pathways are entirely independent.

Comment 5: The introduction suggests that inter-individual factors, such as drinking norms, are more predictive of BD but not necessarily AUD. However, given that social influences often persist in university environments and influence other subdimensions of AUD, such as alcohol-related problems, this assertion may warrant further support or nuanced discussion.

Comment 6: While several studies are referenced, a broader citation base for the dual-pathway model's antecedents, including existing models that address AUD progression, would improve the literature review. Including models that also segment AUD risk factors in a population-specific context (e.g., young adults or university students) could also enhance relevance.

Comment 7: The generalisability (external validity) of the study results could be explained more

Comment 8: The introduction briefly mentions that the findings could inform prevention strategies, but this point could be expanded. What types of interventions might be designed based on a dual-pathway model? Addressing this could provide a more immediate real-world application of the study’s anticipated findings.

Reviewer #2: REVIEW OF PLOS ONE MANUSCRIPT(PONE-D-24-39096]

Title: Predictors of Alcohol Use Disorder Risk in Young Adults: Direct and Indirect Psychological Paths through Binge Drinking

Minor Changes:

Suggested Rephrasing and Consistency: Many sections mentions "Prevention of Alcohol Use Disorder (AUD)”. The psychological factors and determinants being studied are risk factors or predictors for AUD. To the reader, the term "prevention" in this context, may imply that interventions can outright prevent AUD. Instead, emphasizing mitigating or modifying the AUD risk would be a preferable alternative just as the conclusion accurately states that identifying these factors enables risk modification or reduction rather than outright prevention.

Introduction:

Page 5: Similarly, Rephrasing of the last paragraph that "Prevention of Binge Drinking (BD) is tantamount to preventing AUD." This statement oversimplifies the relationship. Instead, emphasize that while reducing BD may mitigate AUD risk, other psychological factors also play a significant role to AUD risk. The above point suggested has been mentioned as concluding statement in the conclusion as well.

Page 5: Inclusion of references to previous studies, especially to discuss the limitations of past research on this topic would be helpful.

Materials and Methods

-Procedures and Participants:

Page 8: Eligibility/Selection Criteria: Authors may expand on the eligibility and inclusion criteria of the sample selected for the study. This will help readers better interpret the study's findings and understand its applicability.

Statistical Analysis:

Page 14: Typological and Grammatical error. Repetition of sentences.

Result: The tables provided are comprehensive and effectively summarize the key findings. However, a mediational diagram or model illustrating the relationships between psychological predictors, binge drinking as a mediator, and AUD outcomes could aid in understanding the mediation pathways. It may be included to enhance the presentation of the results.

Limitations and Future Directions:

- Page 20: Typological error- “Econd” instead of Second.

- Socio-Demographic Insights: Acknowledge the socio-demographic imbalance, particularly the higher percentage of female respondents (67.2%) and discuss potential implications of this gender distribution on the findings.

6. PLOS authors have the option to publish the peer review history of their article (what does this mean? ). If published, this will include your full peer review and any attached files.

**Do you want your identity to be public for this peer review?** For information about this choice, including consent withdrawal, please see our Privacy Policy .

Reviewer #1: No

Reviewer #2: No

---

## [Author Response · Author response to Decision Letter 1]

15 Jan 2025

Dear Editor, Dear Reviewers,

We greatly appreciate your interest for our manuscript, as well as your feedback and comments. We have fully addressed all your comments, leading to an improved version of the manuscript. Please find in the "Response to reviewers" file our specific response to each comment and the changes made to the revised manuscript (in red).

We thank you for your input and hope you will be satisfied with our responses and modifications.

Yours sincerely,

The authors.

---

## [Decision Letter · Decision Letter 1]

14 Mar 2025

Predictors of Alcohol Use Disorder Risk in Young Adults: Direct and Indirect Psychological Paths through Binge Drinking

PONE-D-24-39096R1

Dear Dr. Mauduy,

We’re pleased to inform you that your manuscript has been judged scientifically suitable for publication and will be formally accepted for publication once it meets all outstanding technical requirements.

Kind regards,

Sujiv Akkilagunta, M.D. Community Medicine

Academic Editor

PLOS ONE

Additional Editor Comments (optional):

Reviewers' comments:

Reviewer's Responses to Questions

**Comments to the Author**

1. If the authors have adequately addressed your comments raised in a previous round of review and you feel that this manuscript is now acceptable for publication, you may indicate that here to bypass the “Comments to the Author” section, enter your conflict of interest statement in the “Confidential to Editor” section, and submit your "Accept" recommendation.

Reviewer #1: All comments have been addressed

Reviewer #2: (No Response)

Reviewer #3: All comments have been addressed

Reviewer #4: All comments have been addressed

2. Is the manuscript technically sound, and do the data support the conclusions?

Reviewer #1: Yes

Reviewer #2: Yes

Reviewer #3: Yes

Reviewer #4: Yes

3. Has the statistical analysis been performed appropriately and rigorously? 

Reviewer #1: Yes

Reviewer #2: Yes

Reviewer #3: Yes

Reviewer #4: Yes

4. Have the authors made all data underlying the findings in their manuscript fully available?

Reviewer #1: (No Response)

Reviewer #2: Yes

Reviewer #3: Yes

Reviewer #4: Yes

5. Is the manuscript presented in an intelligible fashion and written in standard English?

Reviewer #1: Yes

Reviewer #2: Yes

Reviewer #3: Yes

Reviewer #4: Yes

6. Review Comments to the Author

Reviewer #1: (No Response)

Reviewer #2: The authors have adequately addressed all the questions and suggestions raised in the previous review. The manuscript is well-structured and provides valuable insights into the study's findings.

However, I have a few observations that could further enhance the clarity and precision of the paper:

Introduction: While comprehensive, the Introduction appears somewhat lengthy. A more concise and lucid presentation would improve readability and engagement, while retaining its essential context.

Terminology – "Pathway" vs. "Relationship": Given that this is a cross-sectional study, could the authors clarify whether the term "relationship" might be more scientifically appropriate than "pathway", as the latter often implies a directional or causal inference that cross-sectional designs cannot establish? While mediation models can be used to study associations between variables, they do not confirm causal pathways with certainty. Could the authors explain their choice of terminology?

Overall, the study is well-conducted, and these refinements would enhance the manuscript’s clarity and rigor.

Reviewer #3: The paper has addressed all the comments by the reviewers.

Comment 1

Thank you for adding Figure 3 . It is well-constructed and adds valuable visual support to the results. However, I recommend explicitly referring to Figure 3 in the results section to ensure readers can easily connect the figure with the corresponding findings.

Reviewer #4: A well written article.

Comment 1:

In the table 1 of Page 12, the total number included for analysis should be mentioned in the table heading. i.e. Sample characteristics (N=2026), so that the readers are clear about the reference number.

Comment 2:

The statistical analysis methods of the paper is well written. Appropriate methods like Ordinal Coefficient alpha is used for the Likert scale. The authors can mention in brief the reason for choosing this method (Advantage) over the traditional Cronbach's alpha for the calculation of internal consistency. This can be mentioned in the Measures section of page 13.

Comment 3:

In Page 17, a brief summary of the figure 3 can be written to explain the key findings and make the readers understand the figure better. It is explained in detail in the following pages 19 and 21. Still a brief 4-5 line summary of all those 3 AUDIT dimensions will give a better understanding.

7. PLOS authors have the option to publish the peer review history of their article (what does this mean? ). If published, this will include your full peer review and any attached files.

**Do you want your identity to be public for this peer review?** For information about this choice, including consent withdrawal, please see our Privacy Policy .

Reviewer #1: No

Reviewer #2: No

Reviewer #3: **Yes: ** AISWARYA LAKSHMI N R

Reviewer #4: **Yes: ** Nishaant Ramasamy

---

## [Editor Report · Acceptance letter]

PONE-D-24-39096R1

PLOS ONE

Dear Dr. Mauduy,

I'm pleased to inform you that your manuscript has been deemed suitable for publication in PLOS ONE. Congratulations! Your manuscript is now being handed over to our production team.

Kind regards,

on behalf of

Dr. Sujiv Akkilagunta

Academic Editor

PLOS ONE